# Functional Annotation Genome Unravels Potential Probiotic *Bacillus velezensis* Strain KMU01 from Traditional Korean Fermented Kimchi

**DOI:** 10.3390/foods10030563

**Published:** 2021-03-09

**Authors:** Sojeong Heo, Jong-Hoon Kim, Mi-Sun Kwak, Moon-Hee Sung, Do-Won Jeong

**Affiliations:** 1Department of Food and Nutrition, Dongduk Women’s University, Seoul 02748, Korea; hsjeong32@hanmail.net; 2The Department of Bio and Fermentation Convergence Technology, Kookmin University, Seoul 02707, Korea; jh9261@naver.com (J.-H.K.); mskwak@kookmin.ac.kr (M.-S.K.); 3KookminBio Corporation, Seoul 02826, Korea

**Keywords:** *Bacillus velezensis* strain KMU01, kimchi, probiotic, starter, γ-glutamyl transpeptidase

## Abstract

*Bacillus velezensis* strain KMU01 showing γ-glutamyltransferase activity as a probiotic candidate was isolated from kimchi. However, the genetic information on strain KMU01 was not clear. Therefore, the current investigation was undertaken to prove the probiotic traits of *B. velezensis* strain KMU01 through genomic analysis. Genomic analysis revealed that strain KMU01 did not encode enterotoxin genes and acquired antibiotic resistance genes. Strain KMU01 genome possessed survivability traits under extreme conditions such as in the presence of gastric acid, as well as several probiotic traits such as intestinal epithelium adhesion and the production of thiamine and essential amino acids. Potential genes for human health enhancement such as those for γ-glutamyltransferase, nattokinase, and bacteriocin production were also identified in the genome. As a starter candidate for food fermentation, the genome of KMU01 encoded for protease, amylase, and lipase genes. The complete genomic sequence of KMU01 will contribute to our understanding of the genetic basis of probiotic properties and allow for the assessment of the effectiveness of this strain as a starter or probiotic for use in the food industry.

## 1. Introduction

Kimchi is the generic term for fermented vegetable foods in Korea, which vary depending upon the main ingredients such as kimchi cabbage and radish. Baechu-kimchi is made with kimchi cabbage (*Brassica rapa*) as a major ingredient, and minor ingredients such as carrot, jeotgal (fermented seafood), and seasonings are added. The added minor ingredients depend upon regional and descending recipes. In particular, recipes from the southern region of South Korea often add fish such as *galchi* (hairtail; *Trichiurus lepturus*) and *jogi* (croaker; *Micropogonias undulatus*). For example, Galchi-baechu-kimchi is made with kimchi cabbage, galchi, and other minor ingredients. Most bacterial studies of kimchi have focused on baechu-kimchi without fish and demonstrated that lactic acid bacteria of the genera *Lactococcus, Lactobacillus*, *Leuconostoc*, *Pediococcus*, and *Weissella* predominate during kimchi fermentation [1,2]. Culture-independent methods also showed lactic acid bacteria as the dominant bacteria along with *Bacillus* in kimchi fermentations [3,4,5]. *Bacillus* strains with antimicrobial activity against multidrug-resistant bacteria have been isolated from kimchi [6]. *B. subtilis* was also isolated from kimchi as a probiotic candidate [7].

*Bacillus* species are spore-forming and ubiquitous bacteria. The spore-forming properties of *Bacillus* confer the advantages of being able to suffer extreme conditions such as sterilization in food processing and the highly acidic conditions of the stomach [8]. Several *Bacillus* strains have been used commercially as probiotics for humans and animals [9]. In addition, *Bacillus* was reported to possess amylase and proteolytic activity, as well as antibacterial activity [10,11]. Due to these enzymatic activities, *Bacillus* has been used as starter cultures for protein, and/or amylase-rich fermented food such as fermented soybean and contributes to the sensory properties of the final products [12,13]. These results suggest the possibility of the organism as a useful starter candidate for food fermentation as well as a probiotic.

*B. polyfermenticus* strain KMU01 was selected from many Galchi-baechu-kimchi-derived strains as a probiotic strain and was patented in Korea [14]. However, *B. polyfermenticus* has not yet been enrolled on the List of Prokaryotic Names with Standing in Nomenclature [15]. Although the genomic sequencing of *B. polyfermenticus* as a *B. velezensis* variant was reported [16], the classification data were insufficient to identify the properties of *B. polyfermenticus*.

The previously isolated KMU01 produced high amounts of γ-glutamyltranspeptidase (GGT) and showed activity under 5% (*w*/*v*) NaCl pressure as well as in no-salt-added conditions. GGT plays an important role in the synthesis and degradation of glutathione as well as drug detoxification [17]. Moreover, GGT contributes to reducing the bitter tastes via the transpeptidation of amino acids having a bitter taste [18]. Therefore, we assumed that KMU01 was appropriate as a useful starter strain and probiotic in the food industry. However, the genomic data of the KMU01 strain were insufficient for a comprehensive picture of the metabolic pathways involved in the general degradation of glutathione and the transpeptidation of amino acids. Moreover, function as a probiotic candidate has not been proposed for the KMU01 strain isolated as *B. polyfermenticus*. Therefore, in this study, we examined the general metabolism and genetic background of the functionality of strain KMU01. Additionally, the superiority of KMU01 as a probiotic and a starter candidate for the food industry was illuminated through comparative genomic analysis.

## 2. Materials and Methods

### 2.1. Bacterial Strains and Culture Conditions

Strain KMU01 from Galchi-baechu-kimchi was subjected to genomic analysis. For experimental proof of the genomic analysis results, strain KMU01 was cultured in tryptic soy broth (TSB; Difco, Detroit, MI, USA) at 37 °C for 18 h to maintain the bacterial traits.

### 2.2. Genome Sequencing

Genomic DNA of strain KMU01 was purified according to the previous study [19], and whole-genome sequencing of strain KMU01 was performed using the PacBio RSII platform (Pacific Bioscience, Menlo Park, CA, USA) with the single-molecule real-time (SMRT) sequencing system (20 kbp) by ChunLab, Inc. (Seoul, South Korea). A total of 11,717,161 reads (228.12 × coverage) were generated, and the reads were assembled into one contig using the CLC Genomics Workbench version 7.5.1 (CLC Bio, Aarhus, Denmark) with the HGAP4 algorithm in SMRT Link (version 7.0.1; Pacific Bioscience). Genome annotation was performed using the NCBI Prokaryotic Genome Annotation Pipeline (version 4.6) [20]. Open reading frames (ORFs) were predicted using Glimmer 3 [21], followed by annotation through a search against the Clusters of Orthologous Groups (COG) database [21].

### 2.3. Comparative Genomics

The genome sequence data from strains B-1 (GenBank Accession No. GCA_000769515.1), LFB112 (GCA_000508265.1), MBE1283 (GCA_001483885.1), and L-H15 (GCA_000833005.1) were obtained from the National Center for Biotechnology Information (NCBI) database (Retrieved 1 November 2020, from http://ncbi.nlm.nih.gov/genomes) for comparative genomic analysis. The average nucleotide identity (ANI) was used to check the similarity of the core genome [22]. Core-genome and pan-genome analyses were performed using the Efficient Database framework for comparative Genome Analyses using BLASTP score Ratios (EDGAR) [23]. Rapid Annotation using Subsystem Technology [24] and the Interactive Pathways Explorer v3 (Retrieved 1 December 2020, from https://pathways.embl.de/) were used to determine the gene contents based on functional subsystem classifications and estimate the amino acid metabolic pathways.

### 2.4. Multilocus Sequence Typing and Phylogenetic Analysis

Multilocus sequence typing (MLST) of the KMU01 strain was performed according to the previous published *Bacillus* MLST scheme using eight housekeeping genes: *adk* (encoding adenylate kinase), *ccpA* (catabolite control protein A), *glpF* (glycerol facilitator), *gmk* (guanylate kinase), *ilvD* (dihydroxy-acid dehydratase), *pur* (phosphoribosyl aminoimidazolecarboxamide formyltransferase), *spo0A* (stage 0 sporulation protein A), and *tpi* (triosephosphate isomerase) [25]. The internal partial sequences of the eight genes were combined manually in the order of *adk*, *ccpA*, *glpF*, *gmk*, *ilvD*, *pur*, *spo0A*, and *tpi* using LaserGene 7.1 software (DNASTAR, Madison, WI, USA). Phylogenetic analysis of the combined sequences was performed using the maximum likelihood method in MEGA 7 [26]. Bootstrapping values were estimated from 1000 repeated calculations. The reference strain gene sequences were obtained from the NCBI.

### 2.5. PCR for the Identification of Toxin Determinants

The amplification of toxin genes from genomic DNA of strain KMU01 was performed with the specific primer sets designed by Abbas et al. [27] using Inclone™ *Taq* polymerase (Inclone Biotech, Daejeon, Korea) in a T-3000 Thermocycler (Biometra, Gottingen, Germany). The samples were preheated for 5 min at 95 °C and then amplified using 30 cycles of 1 min at 95 °C, 30 s at 57 °C, and 1 min at 72 °C. PCR products were migrated in 1.2% agarose gel stained with ethidium bromide and checked by ultraviolet light. *Bacillus cereus* KCCM 11341 was used as a positive control. The experiments were conducted at least three times.

### 2.6. Hemolytic Activity Tests

Tryptic soy agar (TSA; Difco) supplemented with 5% (*v*/*v*) rabbit blood (MB Cell, Seoul, Korea) or 5% (*v*/*v*) sheep blood (MB Cell, Korea) was used for α- and β-hemolytic activity tests, respectively. Strain KMU01 was cultured on TSA containing blood at 37 °C for 24 h, and then the α-hemolytic activity was determined by the formation of clear lytic zones. The β-hemolytic activity was determined by cold shock at 4 °C for 24 h after incubation at 37 °C for 24 h [28]. *Staphylococcus aureus* USA300-P23 and *S. aureus* Newman were used as a positive control for the hemolytic analyses [29]. The experiments were conducted at least three times on separate days.

### 2.7. Antibiotic Susceptibility Test

Antibiotic susceptibility was determined using the disk-diffusion method on Muller–Hinton agar at 30 °C for 24 h based on the guidelines of the Clinical and Laboratory Standards Institute [30]. Eight types of antibiotic disks containing chloramphenicol (30 μg), clindamycin (10 μg), erythromycin (15 μg), gentamicin (30 μg), lincomycin (15 μg), streptomycin (300 μg), tetracycline (30 μg), and vancomycin (30 μg) were purchased from Oxoid (Basingstoke, Hants, UK).

### 2.8. Growth Monitoring in the Presence of NaCl

Strain KMU01 cultured in TSB was normalized to 0.5 at OD_600_ and then diluted 100 times in TSB supplemented with NaCl at concentrations of 5%, 10%, 15%, 20%, and 25% (*w*/*v*) to determine the salt tolerance. Cell growth was monitored for 56 h by measuring the OD_600_ using a Synergy HTX Reader (BioTek, Winooski, VT, USA). All experiments were conducted three times on independent samples prepared in the same way on separate days.

### 2.9. Determination of Antibacterial Activity

The antibacterial activities of strain KMU01 against seven foodborne pathogenic bacteria, *Bacillus cereus* KCCM 11341, *Listeria monocytogenes* ATCC 19111, *Staphylococcus aureus* ATCC 12692, *Alcaligenes xylosoxidans* KCCM 40240, *Flavobacterium* sp. KCCM 11374, *Escherichia coli* O157:H7 EDL 933, *Vibrio parahemolyticus* KCTC 2729, and *Salmonella enterica* KCCM 11862, were determined using the disk-diffusion method on TSA agar at 37 °C for 18 h. Pathogens from an overnight culture in TSB (Difco) were inoculated at 1% (*v*/*v*) in fresh TSB media and incubated to an OD_600_ of 1.0, and then poured onto TSA (Difco).

### 2.10. Determination of Salt Tolerance, Acid Production, and Enzymatic Activity

Acid production was determined on TSA supplemented with 0.7% (*w*/*v*) CaCO_3_. Protease and lipolytic activities were determined on TSA containing 2% (*w*/*v*) skim milk, and tributyrin agar (Sigma-Aldrich, St. Louis, MO, USA) containing 1% (*v*/*v*) tributyrin, respectively. The enzymatic activity and acid production ability were determined by clear halo formation around the colony on each appropriate agar medium after incubation at 37 °C for 18 h. The effect of NaCl on each activity was determined by the addition of NaCl to each medium up to a final concentration of 6% (*w*/*v*).

### 2.11. Nucleotide Sequence Accession Number

The complete genome sequence of *B. velezensis* KMU01 was deposited in DDBJ/ENA/GenBank under the accession number CP063768 and the Korean Collection for Type Cultures under the accession number KCTC 11751BP.

## 3. Results and Discussion

### 3.1. Genetic Identification of Bacillus velezensis KMU01

Strain KMU01 was isolated as *B. amyloliquefaciens* in 2010 and reclassified as *B. polyfermenticus* in 2018 on the basis of 16S rRNA gene sequencing [14]. Now, the 16S rRNA sequence of strain KMU01 showed 99.9% identity with *B. velezensis* NRRL B-41580^T^ and *B. siamensis* SCSIO 05746 (Figure 1A). To identify the precise species, we analyzed the difference based on MLST, which has been used to discriminate between species showing phylogenetic close relatedness, and previously, we proved its efficient discrimination between *Bacillus* species [25]. According to the previously developed MLST scheme [25], we used eight housekeeping genes and generated a maximum likelihood phylogenetic tree with a concatenated alignment of the eight gene sequences for KMU01 (Figure 1B). As shown in the MLST phylogenetic trees, strain KMU01 was clustered in *B. velezensis* and separated from *B. siamensis*. In addition to the MLST, the genetic trees of the eight whole housekeeping genes for the KMU01 strain were clustered in *B. velezensis* (Appendix A). In addition, the ANI of the KMU01 genomic sequence showed 97.8% and 94.4% similarity with *B. velezensis* and *B. siamensis* KCTC 13613^T^, respectively. Therefore, the KMU01 strain was reidentified as *B. velezensis*.

### 3.2. General Genome Properties of Strain KMU01

The complete genome of strain KMU01 was a circular 3,932,437 bp chromosome and did not possess a plasmid. The GC content was 46.5%. Eighty-six tRNA genes and 27 rRNA genes were identified in the genome of KMU01. The genomic analysis predicted 3781 open reading frames (ORFs), and 3475 genes were functionally assigned to categories based on the COG database. The most abundant COG category was related to amino acid transport and metabolism (306 genes, 8.8%), followed by transcription (262 genes, 7.5%) and carbohydrate transport and metabolism (248 genes, 7.1%). The high proportion of genes involved in protein and carbohydrate utilization indicates that strain KMU01 might participate in degrading a wide range of soybean proteins and carbohydrates.

### 3.3. Safety Properties of Strain KMU01

The European Union Food Safety Authority has introduced the Qualified Presumption of Safety (QPS) system to check the safety of microorganisms for the food and feed chain [31]. Several *Bacillus* species are on the QPS list with the provision that they lack toxigenic activity compared with the same genus as the notorious pathogen *B. cereus* [32]. *B. velezensis* was reclassified from *B. amyloliquefaciens* subsp. *plantarum*, *B. methylotrophicus*, and *B. oryzicola* [33]. Among them, *B. amyloliquefaciens* is on the QPS list, but *B. velezensis* is not yet on the list. These results assumed that *B. velezensis* might be a proper species for QPS status if it satisfied the nontoxigenic activity requirement. In the current study, we analyzed the presence of genes related to safety concerns based on genomic information.

#### 3.3.1. Enterotoxin

First of all, we checked for the existence of seven enterotoxin genes found in pathogenic *B. cereus*: three hemolytic enterotoxin genes (*nblA*, *nblC*, and *nblD*), three nonhemolytic enterotoxin genes (*nheA*, *nheB*, and *nheC*), and enterotoxin T gene (*bceT*). Those toxins cause food poisoning. The genomic analysis of strain KMU01 showed that it did not possess those annotated genes or other toxin-related genes such as α-hemolysin. We checked the existence of those genes by PCR to confirm the genomic analysis, but no genes were amplified (Appendix A). Additionally, we checked the hemolytic activities of strain KMU01 using blood media, and strain KMU01 did not exhibit α- or β-hemolytic activity.

#### 3.3.2. Transferrable Antibiotic Resistance

The presence of transferrable antibiotic resistance is an important consideration for safe food bacteria. Antibiotic resistance can indirectly increase virulence, not directly, via quorum sensing [34]. Furthermore, acquired antibiotic resistance genes may be transferred to other bacteria during fermentation and intestinal commensal bacteria as they pass through the intestine [35,36]. Therefore, it is essential to check for the presence of transferrable antibiotic resistance genes.

Based on the COG functional classification, 14 putative antibiotic resistance genes for lincomycin, tetracycline, and multiple drugs were identified in the KMU01 genome (Table 1). The mechanism of antibiotic resistance could be divided into four categories, comprising the modification of antibiotics, inhibition of antibiotic and target binding, modification of the binding site, and reduction of antibiotic penetration and extrusion of antibiotics [37]. The function of the 14 putative antibiotic resistance genes was annotated as efflux pumps or transporters. Thus, we assumed that those genes belonged to the category of reduction of antibiotic penetration and extrusion of antibiotics. Those genes in the KUM01 genome encoded in the chromosome were not strain-specific genes but species-specific based on comparative genomic analysis with four *B. velezensis* strains, B-1 (GenBank Accession No. GCA_000769515.1), LFB112 (GCA_000508265.1), MBE1283 (GCA_001483885.1), and L-H15 (GCA_000833005.1). Most transferable antibiotic resistance has been linked to plasmid-mediated resistance genes, while most of the annotated antibiotic resistance genes from chromosomes were not linked to phenotypic resistance due to low activity [19,38,39,40,41]. The KMU01 strain did not display phenotypically resistance to eight antibiotics including chloramphenicol, erythromycin, lincomycin, and tetracycline (Figure 2). Therefore, these results suggest that the annotated antibiotic resistance genes in the KUM01 genome encoded chromosomally may not contribute to antibiotic resistance.

### 3.4. Functional Properties of the Strain KMU01 Genome

#### 3.4.1. Survival of Strain KMU01 in High-Salt Conditions

Several *Bacillus* species have been detected as dominant species in high-salt fermented foods and showed enzymatic activity under salt conditions [12,42,43]. Additionally, *Bacillus* species contribute to the enhancement of sensory properties through fermentation as well as in high-salt conditions [12,44]. KMU01 strain showed GGT activity under 5% (*w*/*v*) NaCl conditions [14]. KMU01 exhibited growth on TSA plates supplemented with 10% (*w*/*v*) NaCl (Figure 3). These results indicated that *Bacillus* could not only be grown but also display enzymatic activity under high-salt conditions. Therefore, we analyzed genes related to salt tolerance such as synthesis and transport of organic solutes to explain the mechanism of salt tolerance.

The accumulation of solutes such as glycine betaine, proline betaine, trehalose, and glycerol can be used to demonstrate salt tolerance [45]. The KMU01 genome contained two osmoprotectant uptake systems, OpuA and OpuD, of glycine betaine and proline betaine (Figure 4 and Appendix A). OpuA belongs to the ABC transporter, and the KMU01 genome possessed three subunits, the glycine betaine-binding protein OpuAC, transporter OpuAB, and ATPase OpuAA, while OpuD is a single-component transporter. KMU01 also possessed a trehalose operon involved in the uptake and utilization of trehalose and the glycerol uptake system.

The lipid cardiolipin contributes to high salinity stress adaptation [46,47]. The KMU01 genome contained the synthesis pathway of cardiolipin from glycerol-3-phosphate, which comes from glycerol or glucose (Figure 4). Proline is another compatible solute. KMU01 contained proline biosynthesis genes from glutamate or citrate (Figure 4). Glutamate, one of the compatible solutes, also can be produced from the biosynthesis of citrate. The KMU01 strain also possessed a citrate transporter. These results indicate that strain KMU01 could adapt to salt stress environments by accumulating cardiolipin, glycerol, glycine betaine, glutamate, proline, proline betaine, and trehalose.

#### 3.4.2. Probiotic Properties of Strain KMU01

Probiotics are live microorganisms having health benefits for humans and animals. Thus, probiotic strains must be supported by at least one positive human clinical trial with live organisms in the product at an efficacious dose [48].

To exert probiotic effects in the human gut, probiotics must tolerate the harsh conditions of the gastric tract such as low pH and bile salt concentrations [49]. Bile salt hydrolase (EC 3.5.1.24; cholylglycine hydrolase) catalyzes the conversion of conjugated bile salts into free bile salts and confers the survivability of probiotics in the gastric tract [50]. The KMU01 genome encodes for the bile salt hydrolase (IM712_RS05600) gene. Spore-forming ability also confers bacteria with the ability to withstand extreme conditions such as stomach acid, bile, and limited oxygen. The KMU01 genome encoded genes related to the synthesis of the spore cortex, spore coat, spore wall, spore germination, and small acid-soluble spore proteins (Figure 5 and Appendix A). These results suggested that KMU01 might be able to resist extreme conditions and thus survive to reach the intestinal tract through the gastric tract.

Bacterial adhesin is an important property allowing probiotics to reside in the gut [51]. Fibronectin is a glycoprotein that plays a major role in bacterial adhesion [52]. It is well known that biofilm helps to protect against antibiotics and enzymes, as well as stomach acid. Exopolysaccharide (EPS), a major component of extracellular biofilm, also helps aid in adhesion to the human intestinal mucosa [53]. Traditionally, flagellum has been regarded as a motility organelle, but it also functions as an adhesin [54]. The KMU01 genome had a fibronectin-binding protein gene (IM712_RS15940). Interestingly, the KMU01 genome possessed the operon for biofilm formation and flagellum biosynthesis (Figure 5 and Appendix A). The operon for biofilm formation contains EPS biosynthetic genes and transcriptional regulator, and the operon for flagellum contains more than 20 different protein genes for flagellum biosynthesis as well as a two-component system, CheAB, which involved the adhesion of flagella [55].

After adhesion or colonization, bacteria should survive in the gut by successfully competing with other gut microbes. A bacterial strain with bacteriocin-producing ability has a unique advantage for competing with other bacteria in the gut. KMU01 contained the entire lantibiotic mersacidin operon, including the premersacidin (IM712_RS05205), modification protein (IM712_RS05195), and bacteriocin export protein (IM712_RS05185) genes (Figure 5). The mersacidin operon, a bacteriocin gene cluster, was first identified in *Bacillus* and showed a bactericidal effect on Gram-positive bacteria [56]. The presence of bacteriocins in KMU01 suggests its potential to inhibit pathogenic microbes. Phenotypically, strain KMU01 exhibited to inhibit the Gram-positive strains, *B. cereus, L. monocytogenes,* and *S. aureus*, and the Gram-negative strains, *A. xylosoxidans* and *E. coli* (Table 2). These results validate the antimicrobial activities suggested in the KMU01 genomic data.

Probiotics in the gut could provide nutrients such as essential amino acids to the gut microflora and the host [57]. Eight essential amino acids, valine, threonine, phenylalanine, tryptophan, isoleucine, leucine, methionine, and lysine, were suggested externally. Genomic analysis revealed that KMU01 could be involved in the synthesis of 20 amino acids (Appendix A and Figure 6) and KMU01 could produce all eight essential amino acids from glucose. These results suggest that the KMU01 strain might supply needed nutrients to human hosts through the gut.

Gamma-glutamyl-peptide contributes to reducing chronic inflammation through positive allosteric activation of the calcium-sensing receptor [58]. γ-Glutamyl peptides have also demonstrated beneficial effects in human tissue, including antioxidation, anticancer, and lipid-lowering activities [59]. γ-Glutamyl peptides have been detected in several foodstuffs, including fermented foods [60,61]. In fermented foods, γ-glutamyl peptides are formed from free amino acids by GGT and γ-glutamylcysteine synthetase. The genomic analysis revealed that KMU01 encoded two GGT genes (IM712_RS06900 and IM712_RS14420) involved in the production of γ-glutamyl peptides. γ-Aminobutyric acid (GABA) is a well-known bioactive compound that increases the value of fermented soybean products. The KMU01 strain possessed the *puuD* gene (GABA hydrolase family protein) for GABA production from 2-oxoglutarate (Figure 6). Nattokinase (subtilisin; E.C. 3.4.21.62) has shown pharmacologic effects such as antithrombotic and antihypertensive effects [62]. The KMU01 genome possessed the subtilisin gene (IM712_RS18500; E.C. 3.4.21.62), thus demonstrating probiotic determinants in the KMU01 strain. These results suggest KMU01 as a probiotic strain beneficial for human health through the production of γ-glutamyl peptides by GGT, GABA by *puuD,* and antithrombotic activity by nattokinase.

#### 3.4.3. Strain KMU01 as a Starter Candidate

In addition to probiotic properties, the KMU01 genome showed functional ability as a starter candidate for food fermentation. Starter organisms contribute to the sensory properties of food via macromolecule degradation [63]. Therefore, many studies have identified and isolated starter candidates showing strong exoenzymatic activity such as protease, amylase, and lipase activity [43,64,65].

The *B. velezensis* strain KMU01 exhibited amylase, protease, and lipase activity on TSA supplemented with 6%, 6%, and 4% NaCl, respectively (Figure 3). Strain KMU01 possessed an α-amylase gene (IM712_RS02800; E.C. 3.2.1.1) and five lipase genes (Appendix A). Strain KMU01 also encoded 54 protease or peptidase genes (Appendix A). Amylase, lipase, and protease activity contribute to the unique characteristics of fermented foods via the breakdown of carbohydrates, lipids, and proteins to organic acid, esters, amino acids, aldehydes, amines, and free fatty acids. These molecules affect sensory properties during fermentation. γ-glutamyl peptides suggested as functional molecules also contribute to the sensory properties of foodstuff such as kokumi taste, which contributes to continuity, mouthfulness, and thickness [66]. Strain KMU01 also possessed genes related to the production of inosinate (IMP) and disodium guanylate (GMP) (Figure 6). These molecules along with monosodium glutamate are flavor enhancers. These results indicated that strain KMU01 might be useful as a starter candidate for enhancing sensory properties during fermentation.

## 4. Conclusions

The genomic analysis of *B. velezensis* KMU01 revealed the safety of KMU01 based on the absence of acquired antibiotic resistance and enterotoxin genes. Moreover, the KMU01 genome possessed the determinants for human health such as γ-glutamyltransferase, nattokinase, and bacteriocin production. Moreover, several genes related to the enzyme for starch, protein, and lipid degradation and salt tolerance revealed the possibility as an efficient starter candidate for food fermentation in salt conditions. The complete genomic sequence of *B. velezensis* KMU01 will provide the genetic basis for further comparative and functional genomic analyses and help to develop this strain for use in the food, animal feed, and medical industries.

## Figures and Tables

**Figure 1 foods-10-00563-f001:**
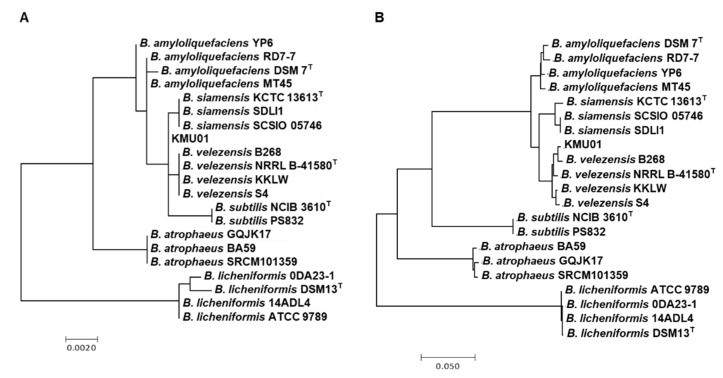
Phylogenetic analysis of 16S rRNA (**A**) and multilocus sequence typing (MLST) based on eight housekeeping genes (**B**). The data were compared using simple matching coefficients and were clustered by the maximum likelihood method. Branches with bootstrap values of 50% have been collapsed. The scale of the diagram is the pairwise distance expressed as the percentage of dissimilarity.

**Figure 2 foods-10-00563-f002:**
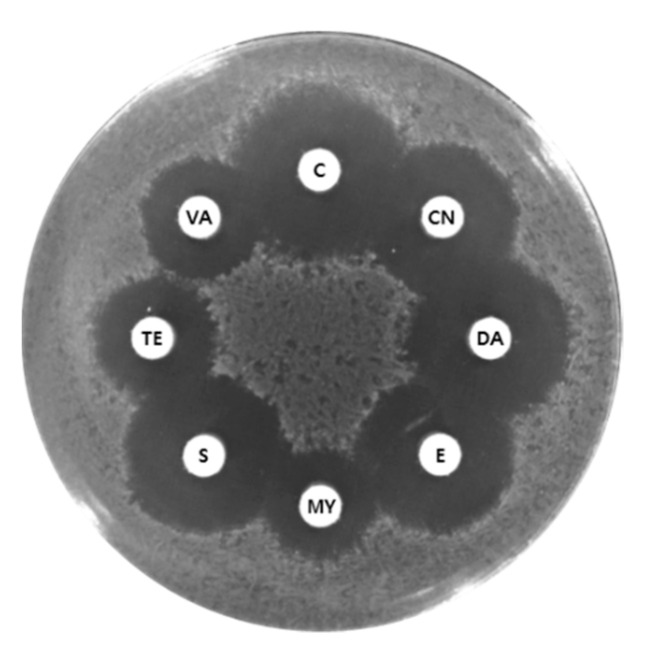
Agar diffusion assay using strain KMU01. Discs: C, chloramphenicol (30 μg); CN, gentamycin (30 μg); DA, clindamycin (10 μg); E, erythromycin (15 μg); MY, lincomycin (15 μg); S, streptomycin (300 μg); Te, tetracycline (30 μg); VA, vancomycin (30 μg).

**Figure 3 foods-10-00563-f003:**
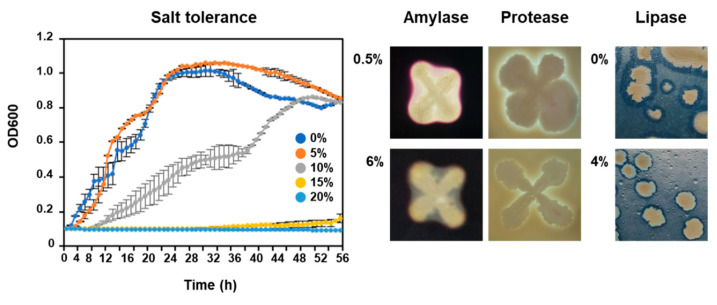
Growth and enzymatic activities of strain KMU01 under salt pressure. Cell growth experiments were conducted three times on independent samples prepared in the same way on separate days.

**Figure 4 foods-10-00563-f004:**
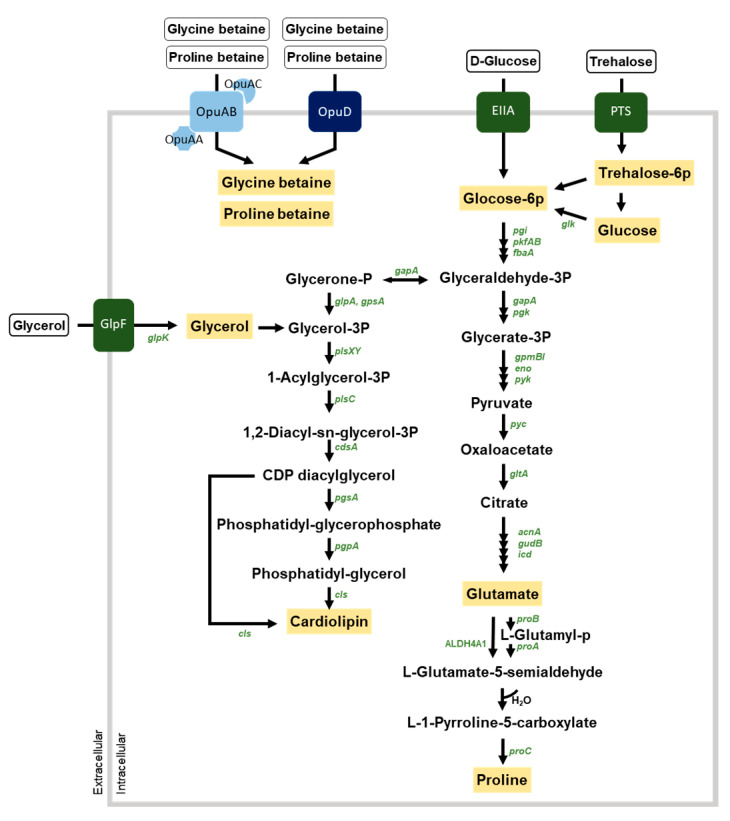
Predicted osmoprotectant transport system and synthesis pathways in strain KMU01. Genes and osmoprotectants are depicted in green italics and letters in blue boxes, respectively. The black arrows correspond to the potential enzymatic reactions catalyzed by gene products encoded in the KMU01 genome.

**Figure 5 foods-10-00563-f005:**
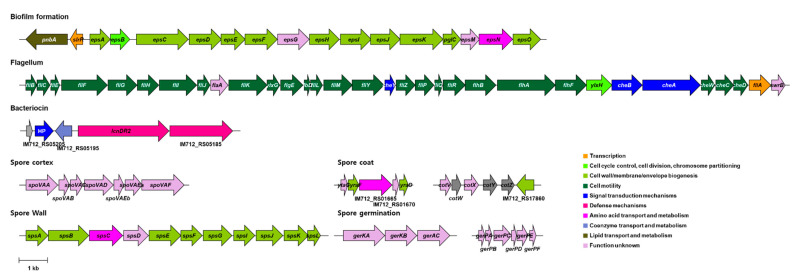
Annotated genes related to the probiotic properties of strain KMU01. The position and orientation of the coding regions are represented by arrows. The name of the arrows is the annotated gene name or the locus number. HP, hypothetical protein.

**Figure 6 foods-10-00563-f006:**
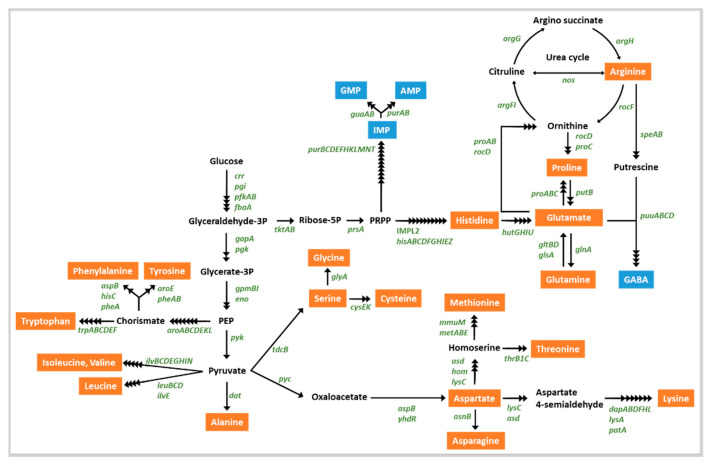
Putative amino acid synthetic pathway in strain KMU01. The names of enzyme-encoding genes are depicted in green. Amino acids and other metabolites are depicted in orange and blue, respectively. The black arrows correspond to potential enzymatic reactions catalyzed by the gene products encoded in the KMU01 genome.

**Table 1 foods-10-00563-t001:** Annotated antibiotic resistance determinants identified in the KMU01 genome and other five *Bacillus velezensis* strains.

Gene Locus	KEGG	Product	COG	Presence of Gene in Five *B. velezensis* Genomes				
				10075	CMT-6	DKU_NT_04	NRRL B-41580	SRCM102741
IM712_RS02140	K03297	Multidrug resistance protein EbrA	P	●	●	●	●	●
IM712_RS02790	K07552	Multidrug resistance protein	P	●	●	●	●	●
IM712_RS02950	K18926	Lincomycin resistance protein LmrB	P	●	●	●	●	●
IM712_RS08810	K08153	Multidrug resistance protein	G	●	●	●	●	●
IM712_RS11555	K08168	Tetracycline resistance protein	G	●	●	●	●	
IM712_RS14095	K03327	Probable multidrug resistance protein NorM	V	●	●	●	●	●
IM712_RS14445	K03327	Probable multidrug resistance protein YoeA	V	●	●	●	●	●
IM712_RS15085	K11814	Multidrug resistance protein EbrA	P	●	●	●	●	●
IM712_RS15090	K11815	Multidrug resistance protein EbrB	P	●	●	●	●	●
IM712_RS17200	K18925	Multidrug resistance protein YkkD	P	●	●	●	●	●
IM712_RS17205	K18924	Multidrug resistance protein YkkC	P	●	●	●	●	●
IM712_RS18380	K08153	Multidrug resistance protein	G	●	●	●	●	●
IM712_RS18790	K06147	Probable multidrug resistance ABC transporter ATP binding	V	●	●	●	●	●
IM712_RS18795	K06147	Probable multidrug resistance ABC transporter ATP binding	V	–	●	●	●	●

The Kyoto Encyclopedia of Genes and Genomes (KEGG) number is endowed by the KEGG Orthology database (http://www.genome.jp (accessed on 1 December 2020) to unify the representation of gene and gene product attributes across all species. The Clusters of Orthologous Group (COG) categorization was generated by annotated gene functions. Abbreviations: ●, identified; –, not identified.

**Table 2 foods-10-00563-t002:** Antibacterial activity of strain KMU01 against food pathogens.

Indicator Strain		Anti-Bacterial Activity
Gram-positive		
	*Bacillus cereus* KCCM 11341	++
	*Listeria monocytogenes* ATCC 19111	++
	*Staphylococcus aureus* ATCC 12692	++
Gram-negative		
	*Alcaligenes xylosoxidans* KCCM 40240	+
	*Escherichia coli* O157:H7 EDL 933	+
	*Flavobacterium* sp. KCCM 11374	‒
	*Salmonella enterica* KCCM 11862	‒
	*Vibrio parahaemolyticus* ATCC 17802	+

Antibacterial activity was determined by the agar disk-diffusion method. ++, ≥2 mm of lytic zone; +, 2 mm of lytic zone; ‒, non-lytic zone.

## Data Availability

The data presented in this study are available in the article.

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
