# Peer review of "Functional Annotation Genome Unravels Potential Probiotic Bacillus velezensis Strain KMU01 from Traditional Korean Fermented Kimchi"

_foods, 2021, doi:10.3390/foods10030563_

Round 1

Reviewer 1 Report

Dear Authors, The research methodologies are many and the results justify the authors will. In conclusion section the authors should emphasize some beneficial characteristic of the bacteria studied.

I suggest to accept the paper with minor revisions

point to point

Abstract

Bacillus velezensis strain KMU01 showing γ-glutamyltransferase activity was previously isolated from a traditional Korean fermented vegetable, kimchi, and suggested as an appropriate pro-biotic candidate:

This sentence is misleading since just that activity doesn’t suggest that BV is a probiotic candidate.

pro-biotic…pro-tease: please correct

85-86 Genomic DNA was isolated and purified using a Wizard Genomic DNA Purification Kit

131-132 Genomic DNA of strain KMU01 was extracted using the DNeasy Tissue Kit

It is not clear to me why you have used two different kits.

Line 139: BLAST its reference should be cited

Line 144: sheep blood (MB Cell)…please change in (MB Cell, Korea)

3.2 in this section the verb tense could probably be better at present, since that DNA is deposited…

Line 217:  3,932,437…needs a unit

Line 220: 3.781 and 3,475…please uniform the style

Throughout the MS a single style should be used according to Journal guidelines. Please check and correct

3.3.1: genes name must be in italic

Lines 246-248: the authors must cite at least a supplementary table regarding these findings.

Lines 326-327: This sentence is not clear, please revise

Line 356: should urvive…please correct

Line 368: Table 2…please explain the levels of activity (+ or ++ or -)

Author Response

Abstract

Bacillus velezensis strain KMU01 showing γ-glutamyltransferase activity was previously isolated from a traditional Korean fermented vegetable, kimchi, and suggested as an appropriate pro-biotic candidate:

This sentence is misleading since just that activity doesn’t suggest that BV is a probiotic candidate.

> We revise the sentence (Lines 15-17).

pro-biotic…pro-tease: please correct

> I confuse what to correct.

85-86 Genomic DNA was isolated and purified using a Wizard Genomic DNA Purification Kit

131-132 Genomic DNA of strain KMU01 was extracted using the DNeasy Tissue Kit

It is not clear to me why you have used two different kits.

> We usually use the DNeasy Tissue Kit for genomic DNA extraction, but custom service laboratory for genomic sequencing usually used the Wizard Genomic DNA Purification Kit. Therefore, we described the different kits. In revised manuscript, we condense the sentence (Lines 84-89, 127-128). 

Line 139: BLAST its reference should be cited

> Bacillus cereus genome as a reference was used for BLAST. But, in current experiment, we did not obtained amplified fragments. So, we revise the sentence (Lines 136-137).

Line 144: sheep blood (MB Cell)…please change in (MB Cell, Korea)

> We add country (Line 144).

3.2 in this section the verb tense could probably be better at present, since that DNA is deposited…

> Thanks for your comments. But, we thought it was better at past tense to describe the results in whole manuscript.

Line 217:  3,932,437…needs a unit

> We add unit (Line 218).

Line 220: 3.781 and 3,475…please uniform the style

> We revise the sentence (Line 221).

Throughout the MS a single style should be used according to Journal guidelines. Please check and correct

3.3.1: genes name must be in italic

> We revise in italic (Lines 243-245).

Lines 246-248: the authors must cite at least a supplementary table regarding these findings.

> We suggest the supplementary table (Line 249).

Lines 326-327: This sentence is not clear, please revise

> We delete the sentence (Lines 330-332).

Line 356: should urvive…please correct

> We revise the word (Line 361).

Line 368: Table 2…please explain the levels of activity (+ or ++ or -)

> We add the description (Lines 376-377).

Reviewer 2 Report

Manuscript ID : foods-1134563

Title : Functional Annotation Genome Unravels Potential Probiotic Bacillus velezensis Strain KMU01 from Traditional Korean Fermented Kimchi

Reading the article “Functional Annotation Genome Unravels Potential Probiotic Bacillus velezensis Strain KMU01 from Traditional Korean Fermented Kimchi» was of great interest. I consider that the paper subject is in accordance with the scope of the journal Foods. The article is of interest regarding the importance of identifying new probiotic strains with food and health applications. Generally, through the manuscript, you should give more information below the figures (legends are missing) and you should further discuss the data obtained.  The statistical methods are not described at all and should be added in the material and methods part.

Some additional remarks regarding each paragraph are detailed below:

Introduction

L 59-64 : The sentence is not clear. The term “previously” makes the sentence confusing regarding the strain you are talking about. Please clarify.

L71 : What do you mean saying that “genomic data were insufficient for a comprehensive picture of the cellular components » ? What does « cellular components » mean to you ? Please be more precise.

Materials and methods

Paragraph 2.8 : Growth monitoring. How long was the experiment? What CFU.mL-1 or which OD600 was used at the beginning of the experiment?

Paragraph 2.9 : I would say “Determination of antibacterial  activity » instead of “Determination of bacteriocin activity ». Because you never evidenced the synthesis of bacteriocin, I would stay careful.

Results and discussion

Paragraph 3.2 : Discussion on the genomic characteristics is missing. You could compare the values obtained for this genome with those of other Bacillus for example. This paper is based on a genomic study. You should give more genomic and comparative genomic informations.

L225 : Please replace protein by proteins.

L 231-233 : Please make this sentence clearer.

L 246-247 : No toxin-related genes were amplified using PCR. Did you use positive PCR controls? What are they? You need to mention the PCR controls used in the material and methods. Without this kind of controls, you can’t say that you did not amplify any gene.

Table 1 : Please replace KEEG by KEGG in the table.

L294 : “Figure 3-A”. There is no A and B in the figure, please make the modification and precise which part of the text is related to the different parts of the figure.

Figure 3 : Could you explain how statistics were made on the graph? Please add a legend below the figure.

L 326-327 : the sentence concerning the in vitro tests used to study probiotic potential of strains is over simplistic and uncomplete. You should add references.

L 356 : please replace “urvive” by “survive”.

Table 2 : How do you quantify and choose between + and ++? Please add protocol details in the materials and methods and in the legend of the table.

L415 : “ up to 6%, 4% and 6% NaCl”. Please correct. Could you justify the use of the concentrations used. Are they in correlation with physiological conditions?

Conclusion

L 430-431 : “The genomic analysis of B. velezensis KMU01 revealed … the phenotypic results ». This sentence is confusing, please clarify. What kind of scientific data do you want to highlight ?

L433-435 : The prospects are quite limited.

The conclusion needs a rewriting to make it clearer and more attractive. You should clarify the conclusions on your data and improve the research prospects.

In my opinion, this article needs a work of revision, including a description in more details of some experimental conditions, the add of legends with the figures and an improvement of the conclusion part. Moreover, the discussion could be fleshed out; genomes comparison for example could improve the paper. The article is interesting in terms of public health and food industry via the characterization of a new potentially probiotic strain.

Author Response

Title : Functional Annotation Genome Unravels Potential Probiotic Bacillus velezensis Strain KMU01 from Traditional Korean Fermented Kimchi

Reading the article “Functional Annotation Genome Unravels Potential Probiotic Bacillus velezensis Strain KMU01 from Traditional Korean Fermented Kimchi» was of great interest. I consider that the paper subject is in accordance with the scope of the journal Foods. The article is of interest regarding the importance of identifying new probiotic strains with food and health applications. Generally, through the manuscript, you should give more information below the figures (legends are missing) and you should further discuss the data obtained.  The statistical methods are not described at all and should be added in the material and methods part.

Some additional remarks regarding each paragraph are detailed below:

> Thanks for your comments.

Introduction

L 59-64 : The sentence is not clear. The term “previously” makes the sentence confusing regarding the strain you are talking about. Please clarify.

> We revise the sentence (Line 57).

L71 : What do you mean saying that “genomic data were insufficient for a comprehensive picture of the cellular components » ? What does « cellular components » mean to you ? Please be more precise.

> We revise the sentence (Lines 69-70).

Materials and methods

Paragraph 2.8 : Growth monitoring. How long was the experiment? What CFU.mL-1 or which OD600 was used at the beginning of the experiment?

> We revise the sentence (Lines 161-164).

Paragraph 2.9 : I would say “Determination of antibacterial  activity » instead of “Determination of bacteriocin activity ». Because you never evidenced the synthesis of bacteriocin, I would stay careful.

> We revise the sentence (Line169).

Results and discussion

Paragraph 3.2 : Discussion on the genomic characteristics is missing. You could compare the values obtained for this genome with those of other Bacillus for example. This paper is based on a genomic study. You should give more genomic and comparative genomic informations.

> We agree to the reviewer’s opinion and we discussed the specific genes in other paragraph.

L225 : Please replace protein by proteins.

> We revise the sentence (Line 226).

L 231-233 : Please make this sentence clearer.

> We revise the sentence (Lines 232-234).

L 246-247 : No toxin-related genes were amplified using PCR. Did you use positive PCR controls? What are they? You need to mention the PCR controls used in the material and methods. Without this kind of controls, you can’t say that you did not amplify any gene.

> We used Bacillus cereus KCCM11341 as a positive control and supplementary table were added (Lines 139 and 249).

Table 1 : Please replace KEEG by KEGG in the table.

> We revise the word in table 1.

L294 : “Figure 3-A”. There is no A and B in the figure, please make the modification and precise which part of the text is related to the different parts of the figure.

> We revise the sentence (Line 297).

Figure 3 : Could you explain how statistics were made on the graph? Please add a legend below the figure.

> We add the legend for explanation of the error bar (Lines 302-304).

L 326-327 : the sentence concerning the in vitro tests used to study probiotic potential of strains is over simplistic and uncomplete. You should add references.

> We delete the sentence (Lines 330-332).

L 356 : please replace “urvive” by “survive”.

> We revise the sentence (Line 361).

Table 2 : How do you quantify and choose between + and ++? Please add protocol details in the materials and methods and in the legend of the table.

> We add the description (Lines 376-377).

L415 : “ up to 6%, 4% and 6% NaCl”. Please correct. Could you justify the use of the concentrations used. Are they in correlation with physiological conditions?

> We revise the sentence (Lines 416-418).

Conclusion

L 430-431 : “The genomic analysis of B. velezensis KMU01 revealed … the phenotypic results ». This sentence is confusing, please clarify. What kind of scientific data do you want to highlight ?

L433-435 : The prospects are quite limited.

The conclusion needs a rewriting to make it clearer and more attractive. You should clarify the conclusions on your data and improve the research prospects.

In my opinion, this article needs a work of revision, including a description in more details of some experimental conditions, the add of legends with the figures and an improvement of the conclusion part. Moreover, the discussion could be fleshed out; genomes comparison for example could improve the paper. The article is interesting in terms of public health and food industry via the characterization of a new potentially probiotic strain.

> We revise the sentence (Lines 432-439).